# Accelerating prediction of chemical shift of protein structures on GPUs: Using OpenACC

**Eric Wright**[1], **Mauricio H. Ferrato**[1], **Alexander J. Bryer**[2], **Robert Searles**[1], **Juan R. Perilla**[2], **Sunita Chandrasekaran**[1] *

**1** Dept. of Computer and Information Sciences, University of Delaware, Newark, Delaware, United States of America, **2** Department of Chemistry & Biochemistry, University of Delaware, Newark, Delaware, United States of America

* schandra@udel.edu

## Abstract

Experimental chemical shifts (CS) from solution and solid state magic-angle-spinning nuclear magnetic resonance (NMR) spectra provide atomic level information for each amino acid within a protein or protein complex. However, structure determination of large complexes and assemblies based on NMR data alone remains challenging due to the complexity of the calculations. Here, we present a hardware accelerated strategy for the estimation of NMR chemical-shifts of large macromolecular complexes based on the previously published PPM_One software. The original code was not viable for computing large complexes, with our largest dataset taking approximately 14 hours to complete. Our results show that serial code refactoring and parallel acceleration brought down the time taken of the software running on an NVIDIA Volta 100 (V100) Graphic Processing Unit (GPU) to 46.71 seconds for our largest dataset of 11.3 million atoms. We use OpenACC, a directive-based programming model for porting the application to a heterogeneous system consisting of x86 processors and NVIDIA GPUs. Finally, we demonstrate the feasibility of our approach in systems of increasing complexity ranging from 100K to 11.3M atoms.

## Author summary

Nuclear magnetic resonance (NMR) spectroscopy yields chemical shifts (CSs) which reveal chemical details of the environment of an atom in a protein. Computer estimation of CSs require the calculation of several contributing terms including interatomic distances, ring current effects and the formation of hydrogen bonds. Here, taking advantage of graphic processing units (GPUs), the estimation of chemical shifts are accelerated thus enabling the determination of the CSs for large systems, encompassing millions of atoms. The rapid determination of CSs enables the use of CS-based validation for other molecular dynamics computations.

This is a *PLOS Computational Biology* Software paper.

**Data Availability Statement:** For source code, please refer to https://github.com/UD-CRPL/ppm_one. For dataset, please refer to accession code 3J3Q and 3J3Y (rcsb.org).

**Funding:** This material is based upon work supported by the National Science Foundation (NSF) under grant no. 1814609 and 2027096. The work is also supported by the NIGMS and NIAID (P50GM082251 and P30GM110758). This work used the Extreme Science and Engineering Discovery Environment (XSEDE), which is supported by the National Science Foundation (NSF grant OCI-1053575). The funders had no role in study design, data collection and analysis, decision to publish, or preparation of the manuscript.

**Competing interests:** The authors have declared that no competing interests exist.

## Introduction

Computing architectures are ever-evolving. As these architectures become increasingly complex, we need better software stacks that will help us seamlessly port real-world scientific applications to these emerging architectures. It is also important to prepare applications that can be readily retargeted to existing and future systems without the need for drastic code changes while maintaining high performance. However, this is a complex and sometimes an impossible task to accomplish.

Programming and optimizing for different architectures at a minimum often require codes to be written in different programming languages thus needing to maintain an entire secondary code base and presenting an inherent difficulty for software developers. While ideally, a single programming standard is preferred, it comes with challenges: (1) Poorly structured algorithms can hide parallelism from hardware (2) Features in a programming model are often hardware-facing and only occasionally application/user-facing, and (3) Hard to design many levels of abstractions to address all problems under study.

Libraries, languages, and directives are three widely accepted software solutions. Libraries suffer from an inherent scope problem; they can only solve a specific subset of problems and are only designed for a specific subset of architectures. Languages are flawed because of the reasons previously outlined such as requiring the programmer to rewrite significant amounts of code. Directives are hints given to compilers to create necessary executables for the underlying platform. Directives strive to offer portability without losing performance.

OpenMP [1, 2] and OpenACC [3] are two widely popular directive-based models. OpenMP is a shared-memory programming model that started to support heterogeneous computing systems since 2013 (OpenMP 4.0 offloading). Applications using the offloading model include Pseudo-Spectral Direct Numerical Simulation-Combined Compact Difference (PSDNS-CCD3D) [4] and Quicksilver [5]. OpenACC, ratified in 2011, has since been adopted widely by scientific developers, to port their large scientific applications—sometimes production code—to heterogeneous architectures. Some examples include ANSYS [6], GAUSSIAN [7], nuclear reactor code Minisweep [8], and Icosahedral non-hydrostatic (ICON) [9]. Both OpenMP and OpenACC allow incremental improvement to a given code base and help create a re-usable code for more than one architecture.

This manuscript focuses on the OpenACC model. We use the PGI compiler after observing their OpenACC implementation's maturity. GCC's (by Mentor Graphics) also offers an OpenACC implementation, however at the time of running this experiment the implementation was not yet mature enough.

### Overview of the scientific problem: Chemical shift prediction

Nuclear magnetic resonance (NMR) is an experimental technique employed in numerous fields such as chemistry, physics, biochemistry, biophysics and structural biology. A chemical shift, the principle observable in NMR instrumentation, provides valuable insight into protein secondary structure by allowing inference about conformation to be drawn based on peak shift. Measured in parts per million (ppm), a chemical shift describes the resonant frequency of a nucleus by comparing its observed frequency to that of a standard reference in the presence of a magnetic field. Magnetic resonance imaging, or MRI, is a familiar application of this powerful technology.

A central challenge in NMR spectroscopy is the structural determination of large proteins. Biomolecular complexes, such as the protein envelopes, or *capsids*, which enclose and protect retroviral genomes often contain symmetries and comprise numerous repeated subunits creating difficulty in NMR experiments. Solid-state NMR (ssNMR) is a powerful emerging solution,

and has successfully elucidated morphological details of multi-million atom complexes such as the HIV-1 protein capsid [10, 11].

With the growing sample sizes accessible to biomolecular experiments, the capability of software and hardware to process and analyze resulting data is also advancing. A method to calculate a continuum electrotatics model, particularly relevant in computational drug-binding studies, has been applied to a 20 million atom system and demonstrated parallel efficiency of 0.8, requiring less than a minute of wall time with 512 GPUs [12]. In biomolecular applications, this trend of increasing computing power is motivating data-driven solutions to problems such as parameterization of atomic force fields [13], or protein structure determination from electron microscopy data [14].

Computational tools to aid structure determination with NMR observables have materialized into a rich domain of protein study and protein chemical shifts have been used in varying ways to successfully elucidate structure. Commonly, these programs employ perusal of scientific databases to establish and parse relationships between shifts, sequence and structure [15–20]. Thanks to projects such as the BioMagResBank (BMRB) [21], NMR data is more available than ever before, engendering the feasibility of semi-empirical prediction methods which utilize existing chemical shift data to parameterize functional prediction models.

Obviating the need for database searching and sequence matching is a semi-empirical method named PPM [22]. The goal of PPM is to provide a prediction model that could operate over NMR conformational ensembles, predict chemical shifts from structures and provide new dimensions of protein forcefield refinement, structural refinement, and ensemble validation—a goal which PPM met aptly. In a departure from ensemble analysis, PPM's successor PPM_One introduced a static-structure based chemical shift prediction method that showed competitive accuracy with other software [23].

## Motivation

Drawing from approximations of first principle calculations and trained with accessible NMR data, the PPM_One model considers chemical shift as a sum of discrete *descriptors*. These descriptors, which quantify chemical shifts due to ring current effects, hydrogen bond effects, dihedral angles, and more [22, 23], take the form of relatively simple, and differentiable, functions of the atomic coordinates. Considering these factors, PPM_One is a prime target for parallelization and optimization; to extend practical application of the software to larger structures, populous NMR ensembles, or molecular dynamics trajectories describing thousands of structures. While a suitable candidate to this end, the original PPM_One code was not written in a way to exploit the massive compute power of accelerators such as GPUs. In our work, we have ported the PPM_One application to utilize parallel hardware, such as GPUs, using OpenACC.

This work makes the following contributions:

- Equip domain scientists with an accelerated version of PPM_One that functions in a realistic lab environment.

- Provide an accelerated chemical shift prediction code that can be adapted to large Molecular Dynamics packages.

- Demonstrate the feasibility and scalability of our approach in systems of increasing complexity ranging from 2,000 to 13,000,000 atoms.

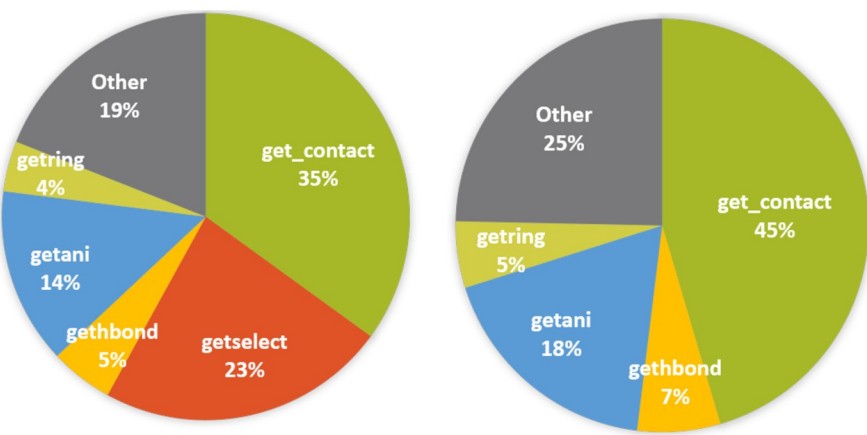

(a) Initial serial profile collected from PGPROF        b) PGPROF reprofile after optimizing serial portions of the code

**Fig 1. Visual representation of serial profiling data.** (A) The pie chart represents the time taken by the original version of the code. (B) The pie chart represents the time consumed by the different parts of the code after implementing various optimizations.

## Design and implementation

This section will discuss methods to determine the computationally intensive hotspots, steps taken to refactor the code, accelerate using OpenACC and incrementally improve the application of OpenACC directives.

### Identifying computational hotspot in chemical shift prediction

Before accelerating or parallelizing a given code, we use the OpenACC-enabled profiler that comes packaged with the PGI compiler. The tool, PGPROF, displays detailed information about CPU and GPU performances. This information includes breakdowns by runtime, memory management, and accelerator utilization. Fig 1 shows the results of our profile when using a relatively small molecule (100,000 atoms). The profiler was particularly useful since we were unfamiliar with the code at the start of this project, and PGPROF quickly allowed us to identify which functions in PPM_One contained a lot of computation as well as which functions scaled in time-taken with the dataset size. The different computational functions detected are discussed in detail in the *Target Functions for Acceleration* section.

### Initial code refactoring

Many of these functions in the original sequential code were written as a direct implementation of their respective algorithms. As a result they are under prepared for accelerators. For example, redundancy of memory copying caused by calling the *getselect*() function an unnecessary number of times. To fix this, we altered the code to only call *getselect*() once, and then store and reuse the associated memory. This optimization alone led to a 20% performance increase when running with some of the datasets.

The next optimization we made was to a function called *clear*() that filters through a list of protons, removing any of them that do not work with the algorithm. The runtime (varying

from hours to seconds) of this function varied greatly depending on which dataset was tested since some molecular structures require more protons to be filtered than others. As a result, we rewrote *clear*() to use a more efficient list filter that made the operation take only a few seconds or less for all structures.

Lastly, we ran into some problems with the C++ STL containers that were used within the code. This mostly applied to the C++ standard vector class. To account for this, many C++ vectors were replaced with basic arrays, this allowed for more efficient communication with the GPU. In other places, we interfaced with the vector containers by using the built-in *data*() function to retrieve the underlying memory, allowing us to move the data to the GPU without the need to use extra libraries or code rewrites.

## Acceleration using OpenACC

OpenACC exposes three levels of parallelism via the gang, worker and vector constructs that enables programmers to abstract the architecture along with maximally utilizing the potential of multicore or accelerators. Typically, compute-intensive portions of the program often identified by profilers are offloaded to the accelerators; a task orchestrated by the host by allocating memory on the accelerator device, initiating data transfer, offloading the code to the accelerator, passing arguments to the compute region, queuing the device code, waiting for completion, transferring results back to the host, and deallocating memory. With often only minor adjustments to memory management near parallelized compute regions, the model accommodates both shared and discrete memory or any combination of the two across any number of devices. The model has the capacity to expose the separate memories through the use of a device data environment.

After ensuring that the code was accelerator-compatible, we began applying OpenACC directives to the code. We tackled each function individually in order of importance, meaning that we started with *get_contact*() and finished with *getring*(). Every time we made a meaningful alteration we would re-run the code on a few different datasets and compare the results to their non-accelerated baselines. This would let us know if we made any errors along the way.

We decorated the major loops in the code with the OpenACC *parallel loop* directive. This will offload loops to the GPU automatically; sometimes just enough to see a speedup as some loops were embarrassingly parallel. However, in other cases we saw a significant slowdown and sometimes wildly incorrect code output compared to our serial baseline. These two problems were overcome by using other OpenACC features.

To fix our incorrect output, we used both the *reduction clause* and *atomic directive*. Reduction clause handles race conditions. These are areas in the code that can result in errors when multiple parallel units overwrite each other in shared memory. The reduction clause prevents this by aligning memory reads/writes to produce a single coherent value.

The atomic directive fills a similar purpose. However, it is useful in situations where many different race conditions could occur at different locations in memory. There was only one situation in our code where a reduction clause was not sufficient, and that was in the *gethbond*() function.

Too many memory transfers between the host and device slowed down the code. After profiling our initial parallelization of the *get_contact*() function, we saw that the majority of the time was spent on transferring data between the host and device memory. Originally, *get_contact*() would be called many times throughout code execution (hundreds to thousands of times, depending on the dataset). We added a loop that would iterate over all of the individual *get_contact*() calls, which gave us another dimension to expose parallelism. This also means that no data would need to be transferred between the different calls of *get_contact*(). This

change was beneficial because out of all of the functions *get_contact*() received the largest speed-up. The speed-up will be discussed in more detail in the Results section.

## Target functions for acceleration

Each of the functions we have identified are important to the overall chemical shift prediction algorithm that PPM_One implements. *get_contact*(), one of the most important functions in the PPM_One algorithm, serves as the principle interface between the input coordinates and secondary structure contact data. *get_contact*() iterates over all atomic positions, given in the molecule, and computes a distance between each atom index and the successive atom index. Next, for each atom in each residue in the PPM_One input structure, the random-coil chemical shift for atoms in that residue is applied as a fit parameter to normalize the calculated chemical shift. Since this procedure must be carried out exhaustively over the entire structure and manages data from individual function calls and parameter tables, it takes up a proportionally large piece of the total runtime and can be a huge sequential bottleneck in the program.

*gethbond*() computes the effect that backbone hydrogen bonding has on chemical shift. PPM_One describes this effect in terms of the inverse of donor-acceptor distance, and applies a descriptor based on the angle formed between two different atom triples, NHO and $HOC'$. Since every amino acid has donor-acceptor pairs, this function gets called with high frequency and involves distance and angle calculations for each donor and acceptor relative to the specified atom triples making *gethbond*() a meaningful target for parallelization and performance-gain despite its relatively simple formulation.

The *getani*() function computes the chemical shift due to magnetic anisotropy. Magnetic anisotropy quantifies the directionally-dependent electromagnetic interactions between atoms. PPM_One employs this calculation for interactions between protons and peptide-amide groups consisting of Oxygen ($O$), Carbon prime ($C'$) and Nitrogen ($N$). Additional calls are made to *getani*() for side-chain $OCN$ groups of Asparagine and Glutamine, $OCO$ side-chain groups of Glutamate and Aspartate, and the $NCN$ side-chain of Arginine. The formulation for the calculation used by PPM_One is known as the "axially symmetric model" [24], in accordance with McConnell's characterization of anisotropy of peptide groups [25]. At each function call, the distance between the queried proton and the peptide-amide group is calculated. This, the vectors pointing from the proton to the peptide-amide group, and from the proton to the normal vector of the peptide-amide are used to compute an angle to pass into the magnetic anisotropy expression.

*getring*() encompasses two different functions in the PPM_One program that calculate the chemical shift due to ring-current effects; one function calculates ring-current effects felt by Hydrogen atoms with respect to an aromatic residue, and the other calculates the effect felt by backbone atoms adjacent to an aromatic residue. PPM_One considers the aromaticity of amino acids Phe, Tyr, His, Trp-5 and Trp-6. The aromatic rings of these residues have important structural implications due to electrostatic induction, as the circular movement of delocalized electrons (ie, current) in conjugated Pi-bonding orbitals induces a magnetic field vector orthogonal to the plane described by the atoms of the ring. To quantify this effect, the queried atom's position in cartesian space must be projected to a position on the 2D subspace defined by the plane of the aromatic ring. Additionally, distances between all atoms in the ring are calculated in this function each time it is called, making it costly to compute even though its application is limited to only aromatic residues and atoms in their local environment.

## Results

This section will elaborate on the experimental setup and the results obtained.

### Experimental setup

For the multicore, V100 and P40 results shown in both the tables, we use the PSG DGX-1b compute node consisting of Intel Xeon e5-2698 v4 20 cores and a single NVIDIA Volta V100 card and another compute node that has a single P40. For the serial runs shown in both the tables, since we could not get time on the PSG system, we have used our internal UDEL's local system that has an Intel x990 core.

### Datasets

Fig 2 shows the different datasets used for our experiments, represented to scale. The first tested dataset constitutes 100,000 atoms, roughly a quarter-turn, of the Dynamin GTPase (structure E) extracted and written to their own Protein Database (PDB) file. Structure B was the HIV-1 capsid assembly (CA) without Hydrogens. This structure was tested without Hydrogens for two reasons: 1) to limit the number of atoms for this test case and 2) to create a variety in the swath of tested structures. Structures C and D correspond to two variants of the HIV-1 CA, Hydrogens included. Structure C is the HIV-1 CA decorated with Cyclophilin A (CypA), structure D is the same HIV-1 CA decorated with Myxovirus resistance protein B (MxB). These two datasets, 5.1 and 5.9 million atoms respectively, were chosen as test cases of heterogeneous systems in addition to their increased atom counts compared to the undecorated HIV-1 CA. The HIV-1 CA test-structures are shown next to their dimeric building block 2KOD (structure A), illustrating the ranging scale and complexity of atomistic representations of biomolecules. Finally, the largest two test systems were built from the Dynamin GTPase. Structure E is a 6.8 million atom model, 14 turns, of the GTPase. The largest structure, containing 13.6 million atoms, constitutes 28 turns of the Dynamin GTPase. The secondary-structure of 2KOD was calculated using Stride [26]. All images were rendered using VMD 1.9.4 and the co-distributed, Tachyon parallel ray-tracing library [27, 28].

When running the PPM_One application, we noticed that the total runtime is proportional to the number of atoms contained in the molecule. However, this is not the only deciding factor. Between the different-sized molecules, the various compute-intensive functions saw a linear runtime increase compared to total number of atoms. However, the data preprocessing that the code does can vary greatly based on the molecule, and while we have made many improvements to this step it is still the bottleneck of the application. Also, the function *gethbond*() will take a significant amount of time for molecules that contain hydrogen, and almost no runtime for molecules that do not contain hydrogen. To accommodate for these runtime differences, we are mostly concerned with performance increase of a molecule on different platforms and less concerned with comparing different molecules to each other.

When observing Table 1 we see a significant decrease in total runtime when comparing the serial (optimized) run to any of the accelerators. The multicore performance was 18x faster than the single core results. The Volta V100 results were 56x faster than single core, and 3.1x faster than multicore.

When observing individual function performance we see more significant speedup numbers as shown in Table 2. Comparing V100 results to the multicore results, the *get_contact*() function was sped up by 258x, *gethbond*() by 11x, *getani*() by 10x and *getring*() by 3x. Such a high speed up is common for functions that are purely compute intensive and hence can be easily optimized for GPUs. Since our major computational functions are seeing this amount of increase, we predict that much of the remaining total runtime is bound by other portions of

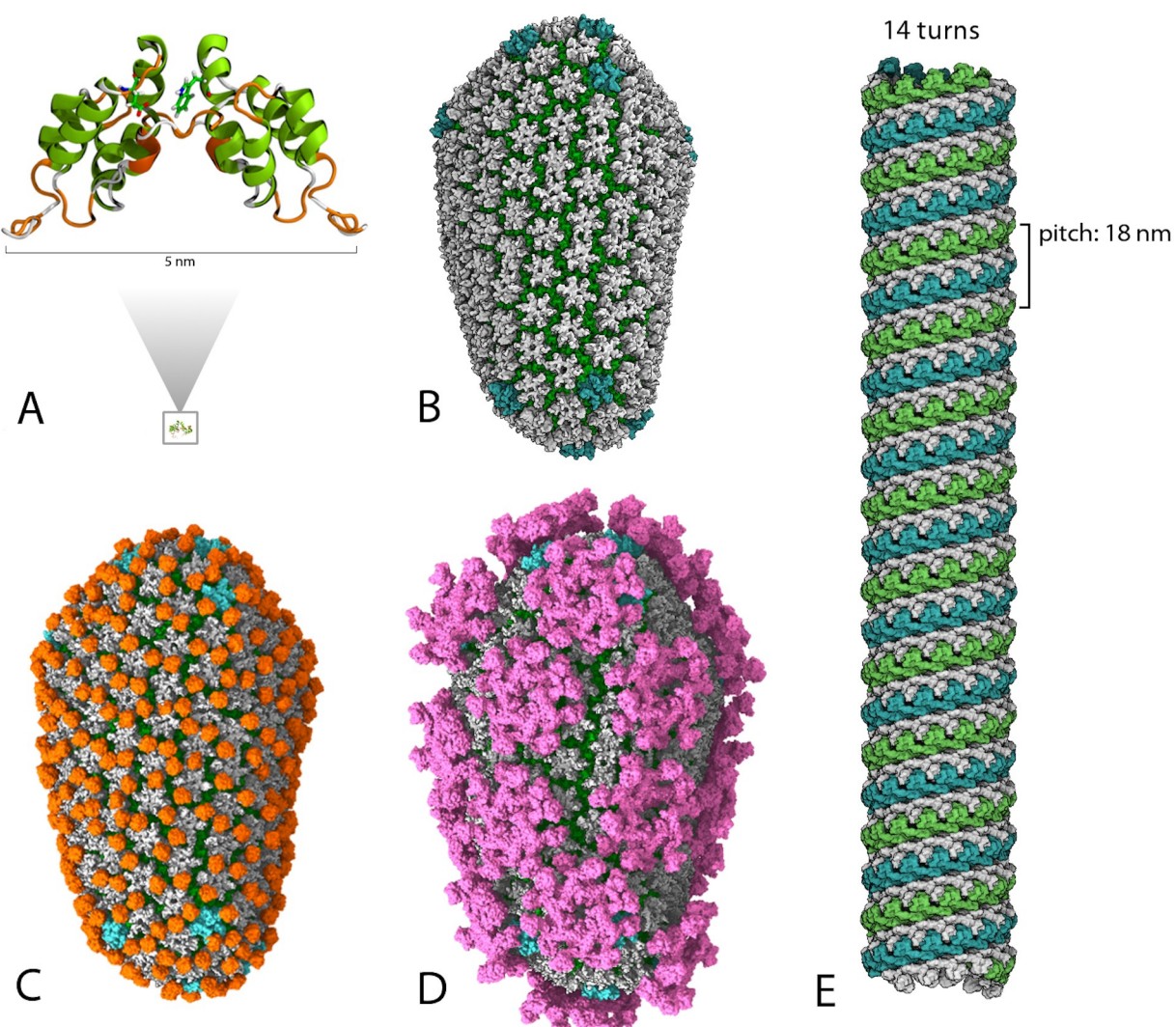

**Fig 2. Visual rendering of used datasets.** (A) The first tested dataset constitutes 100,000 atoms, roughly a quarter-turn, of the Dynamin GTPase extracted and written to their own Protein Database (PDB) file. (B) Structure B was the HIV-1 capsid assembly (CA) without Hydrogens. (C) Structure C is the HIV-1 CA decorated with Cyclophilin A (CypA). (D) structure D is the same HIV-1 CA decorated with Myxovirus resistance protein B (MxB). (E) Structure E is a 6.8 million atom model, 14 turns, of the GTPase.

**Table 1. Results for small to large dataset.**

|  | 100k atoms | 1.5m atoms | 5m atoms | 6.8m atoms | 11.3m atoms |
|---|---|---|---|---|---|
| Serial (Unoptimized) | 167.11s | 572.01s | 3547.07s | 7 hrs (esimate) | 14 hrs (estimate) |
| Serial (Optimized) | 53.57s | 196.12s | 2003.6s | 1510.71s | 2614.4s |
| Multicore | 4.67s | 32.82s | 116.66s | 153.8s | 146.06s |
| P40 | 3.47s | 17.15s | 56.2s | 78.57s | 72.55s |
| V100 | 3.11s | 13.62s | 39.79s | 49.63s | 46.71s |

For these results, an Intel Xeon e5-2698 v4 20 cores CPU and a NVIDIA Volta V100 GPU were used.

**Table 2. Runtime for medium dataset by function.**

| 5m atoms | Total Runtime | get_contact | getani | getring | gethbond |
|---|---|---|---|---|---|
| Serial (Optimized) | 2003.60 | 1177.61s | 58.95s | 22.53s | 708.07s |
| Multicore | 116.66s | 51.73s | 2.4s | 0.6s | 25.39s |
| P40 | 56.2s | 1.69s | 1.06s | 0.5s | 17.05s |
| V100 | 39.79s | 0.2s | 0.24s | 0.18s | 2.35s |

the code such as file I/O or preprocessing. We have improved these parts of the code significantly since the start of this project (as seen when comparing the serial unoptimized numbers against the serial optimized). We do not believe that too much more could be done to improve these aspects without rewriting large portions of the code.

## Validation of results: Calculation RMSE

To calculate the Root Mean Square Error (RMSE), we ran the unaltered code on a single core of a single CPU on 299 different PDB files. Then we reran each file with the developed Open-ACC code on the same CPU core, but now with GPU offloading. The following numbers shown in Table 3 are collected by using the RMSE formula on every prediction of every file comparing the CPU and GPU output.

$$RMSE = \sqrt{\frac{\sum_{i=1}^{n} \left(P_i - O_i\right)^2}{n}} \tag{1}$$

Next, we wanted to assess the prediction accuracy of the PPM_One code against experimentally derived chemical shifts. PPM_One reported root-mean-square prediction error for a set of validation structures [23], showing 0.9 ppm prediction error for Carbon alpha and 1.0 ppm error Carbon beta atoms, 1.41 ppm error for carboxyl Carbon atoms, 0.24 ppm error for Hydrogen alpha and 0.43 ppm for amide Hydrogen atoms, and 2.31 ppm error for Nitrogen atoms [23]. To compare the accuracy of GPU accelerated PPM_One with respect to experimental chemical shifts, chemical shifts were predicted for three structures which were not part of the PPM_One training or validation sets [29–31]. We found comparable root-mean-square prediction error to what was reported for PPM_One [23]: 1.12 ppm prediction error for Carbon alpha and 1.11 ppm error for Carbon beta atoms, 1.03 ppm error for carboxyl Carbon atoms, 0.55 ppm error for Hydrogen alpha and 0.71 ppm for amide Hydrogen atoms, and 1.41 ppm error for Nitrogen atoms.

Together with the RMSE analysis between CPU and GPU versions of the code, we conclude that PPM_One provides robust and accurate chemical shift predictions which were unaffected by our GPU acceleration efforts.

**Table 3. RMSE difference between CPU and GPU code.**

| | C_a | C_b | C | HN | N | H_a |
|---|---|---|---|---|---|---|
| RMS error (ppm) | 1.58e-4 | 8.48e-5 | 1.97e-4 | 5.22e-5 | 2.84e-4 | 1.02e-4 |
| Max error (ppm) | 0.013 | 0.008 | 0.017 | 0.007s | 0.025 | 0.013 |

## Availability and future directions

The PDB files have been previously published and can be found here [32–34]. Our GitHub https://github.com/UD-CRPL/ppm_one contains the code used for this manuscript.

Efficiently predicting chemical shifts is an important utility for many potential MD applications. With our GPU acceleration, we believe that PPM_One can now be used for predicting chemical shifts of large molecular structures. As part of the future work, for problems of magnitude larger than what we have studied, we will update the software to use MPI with OpenACC and scale across multiple nodes.

## Author Contributions

**Conceptualization:** Eric Wright, Mauricio H. Ferrato, Alexander J. Bryer, Robert Searles, Juan R. Perilla, Sunita Chandrasekaran.

**Funding acquisition:** Juan R. Perilla, Sunita Chandrasekaran.

**Investigation:** Eric Wright, Mauricio H. Ferrato, Alexander J. Bryer, Juan R. Perilla, Sunita Chandrasekaran.

**Methodology:** Eric Wright, Mauricio H. Ferrato, Robert Searles.

**Project administration:** Juan R. Perilla, Sunita Chandrasekaran.

**Resources:** Juan R. Perilla, Sunita Chandrasekaran.

**Software:** Eric Wright, Mauricio H. Ferrato, Robert Searles.

**Supervision:** Juan R. Perilla, Sunita Chandrasekaran.

**Validation:** Eric Wright, Alexander J. Bryer.

**Visualization:** Alexander J. Bryer, Juan R. Perilla.

**Writing – original draft:** Eric Wright, Mauricio H. Ferrato, Alexander J. Bryer.

**Writing – review & editing:** Eric Wright, Alexander J. Bryer, Juan R. Perilla, Sunita Chandrasekaran.

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
