## [Decision Letter · Decision Letter 0]

10 Mar 2020

Dear Dr. Chandrasekaran,

Thank you very much for submitting your manuscript "Accelerating Prediction of Chemical Shift of Protein Structures on GPUs" for consideration at PLOS Computational Biology. As with all papers reviewed by the journal, your manuscript was reviewed by members of the editorial board and by several independent reviewers. The reviewers appreciated the attention to an important topic. Based on the reviews, we are likely to accept this manuscript for publication, providing that you modify the manuscript according to the review recommendations.

Sincerely,

Dina Schneidman-Duhovny

Software Editor

PLOS Computational Biology

Dina Schneidman-Duhovny

Software Editor

PLOS Computational Biology

[LINK]

Reviewer's Responses to Questions

**Comments to the Authors:**

Reviewer #1: This work present a systematic report that accelerate the PPM_One program by implementing portable optimization using OpenACC. It includes detailed description of the process helped improved the efficiency of the PPM_One orders of magnitude. It will greatly benefit the NMR communities and structural biology work involving NMR methods. The manuscript is nearly ready for publication, in my opinion. Here are some minor comments:

1. Page 2-3, Section "Overview of the Scientific Problems: Chemical shift Prediction: The citations in the first three paragraphs do not show up corrected. Please correct them.

2. Page 6, the end of paragraph 4: The speed-up will be discussed in more detail in Section 3.2. However, sections are not numbered, no idea which this refers to.

3. Page 7, first sentence of paragraph 2: The function getani() represents the compute region for calculating the chemical shift due to magnetic anisotropy. This sentence may need revision.

4. Table 1. The times for 6.8m atoms are longer than 5m atoms, which is easy to understand. However, why is it longer than 11.3m atoms? Any particular reason? Would love to see the explanation. Or maybe this reveals there is something else need optimization?

Reviewer #2: The manuscript at high level is very well organized and written. This work will definitely help other MD application developers think of using / adapting to programming models such as OpenACC to accelerate their applications, without having to delve into the nitty-gritty's of CUDA.

Minor revisions needed in the abstract and throughout the paper, comments indicated in the pdf (please open using Adobe Reader or similar).

Comments:

1. Be explicit in the abstract that there was major refactoring on the cpu side of the code happened as well before accelerating the code to the GPUs. Just 14hrs to 46secs seems like the code was really really bad, and the comparison is not appropriate. Recommend tweaking the statement a bit to reflect the changes (note: this was clear in the results section but not in the abstract).

2. Some citations are missing, Fig 1 was missing (I think at least, not in the pdf that was uploaded), also recommend running the manuscript through Grammarly (or similar) for grammatical error fixes.

3. Are their plans of running such simulations / modeling on larger systems with a higher (and complex) node count and what do the authors think of as a challenge when running similar workloads with OpenACC + MPI (or OpenMPI) ? Maybe something along these lines in a future work, or ongoing work directions would be an appropriate addition to this manuscript.

4. Recommend outlining the challenges of accelerating and porting such an application using OpenACC in a subsection, this will help other MD app developers to take that leap.

Reviewer #3: The manuscript describes a significant improvement on the run time of a chemical shift prediction program.

Chemical shift prediction is an important area of ongoing research to leverage the large amount of data available from NMR experiments of protein. The manuscript does a very good job of addressing the case of solid-state NMR, which can tackle large protein assemblies. The test models were chosen well and demonstrate the challenge associated with chemical shift prediction.

The results show important acceleration gains achieved, and the RMSD comparison clearly indicates a successful implementation. As such this is a significant contribution to the field.

One minor suggestion is that the authors include a brief comment on how successful PPM_One is when compared to experimental NMR results (if available) for the various chosen models.

**Have all data underlying the figures and results presented in the manuscript been provided?**

Reviewer #1: Yes

Reviewer #2: Yes

Reviewer #3: Yes

PLOS authors have the option to publish the peer review history of their article (what does this mean?). If published, this will include your full peer review and any attached files.

Reviewer #1: No

Reviewer #2: No

Reviewer #3: No
---

## [Editor Report · Decision Letter 1]

15 Apr 2020

Dear Dr. Chandrasekaran,

We are pleased to inform you that your manuscript 'Accelerating Prediction of Chemical Shift of Protein Structures on GPUs: Using OpenACC' has been provisionally accepted for publication in PLOS Computational Biology.

Best regards,

Dina Schneidman-Duhovny

Software Editor

PLOS Computational Biology

---

## [Editor Report · Acceptance letter]

1 May 2020

PCOMPBIOL-D-20-00060R1 

Accelerating Prediction of Chemical Shift of Protein Structures on GPUs: Using OpenACC

Dear Dr Chandrasekaran,

I am pleased to inform you that your manuscript has been formally accepted for publication in PLOS Computational Biology. Your manuscript is now with our production department and you will be notified of the publication date in due course.

With kind regards,

Sarah Hammond
